# Bioinspired Platelet-like Nanovector for Enhancing Cancer Therapy via P-Selectin Targeting

**DOI:** 10.3390/pharmaceutics14122614

**Published:** 2022-11-26

**Authors:** Shengli Wan, Yuesong Wu, Qingze Fan, Gang Yang, Haiyang Hu, Singkome Tima, Sawitree Chiampanichayakul, Songyot Anuchapreeda, Jianming Wu

**Affiliations:** 1Department of Medical Technology, Faculty of Associated Medical Sciences, Chiang Mai University, Chiang Mai 50200, Thailand; 2Department of Pharmacy, The Affiliated Hospital of Southwest Medical University, Luzhou 646000, China; 3School of Basic Medical Sciences, Southwest Medical University, Luzhou 646000, China; 4School of Pharmacy, Southwest Medical University, Luzhou 646000, China; 5Luzhou Key Laboratory of Activity Screening and Druggability Evaluation for Chinese Materia Medica, Southwest Medical University, Luzhou 646000, China; 6Research Center of Pharmaceutical Nanotechnology, Chiang Mai University, Chiang Mai 50200, Thailand

**Keywords:** platelet membrane, lipid nanovector, targeted cancer therapy, curcumin

## Abstract

Cancer is a major threat to the health of humans. Recently, various natural products including curcumin (CCM) have attracted enormous interest for efficacious cancer therapy. However, natural therapeutic agents still encounter certain challenges such as rapid clearance, low bioavailability, and poor tumor targeting. Recently, the platelet membrane (PM) camouflaged nanoparticle has provided a promising solution for cancer targeting therapy. Nevertheless, only limited efforts have been dedicated to systematically explore the mechanism of affinity between PM bioinspired nanoparticles and various tumor cells. Herein, a CCM-encapsulated platelet membrane biomimetic lipid vesicle (CCM@PL) with a size of 163.2 nm, zeta potential of −31.8 mV and encapsulation efficiency of 93.62% was developed. The values of the area under the concentration-time curve and mean residence time for CCM@PL were 3.08 times and 3.04 times those of CCM, respectively. Furthermore, this PM biomimetic carrier showed an excellent affinity against Huh-7, SK-OV-3 and MDA-MB-231 cell lines due to the biomolecular interaction between P-selectin on the PM and tumoral CD44 receptors. In addition, CCM@PL displayed enhanced cytotoxicity compared with free CCM and the synthetic formulation. Overall, our results suggest that this developed PM biomimetic lipid nanovector has great potential for targeted cancer treatment and natural components delivery.

## 1. Introduction

Despite the advanced science and technology, cancer is still a disease burden worldwide [1,2]. Conventional methods for cancer treatment include surgery, chemotherapy, and radiation therapy. Traditional chemotherapy has severe problems such as poor specificity and undesirable toxicity. Nowadays, various natural products are important resources against tumor growth [3,4,5,6]. Curcumin (CCM) is a natural phenolic compound extracted from *Curcuma longa*, possessing a favorable anticancer effect and being non-toxic [7,8,9,10]. However, the therapeutic potentials of CCM are universally limited by the low bioavailability, poor tumor affinity, and restricted cellular uptake [11,12]. Nanodrug delivery systems have attracted tremendous attentions in improving the pharmacokinetics and tumor targeting of chemotherapeutics, including phytochemicals [13,14,15,16,17]. Many studies have focused on the development of CCM-loaded nanocarriers, like micelles, polymeric nanoparticles, silica nanoparticles, and lipid vesicles [7,9]. Particularly, the nano lipid vesicle mainly composed with phospholipid and cholesterol is used as an efficient delivery of CCM to enhance the effect of CCM [8,18]. Besides, nano-vectors can enhance CCM tumor accumulation via the enhanced permeability and retention (EPR) effect. Nevertheless, this passive tumor targeting caused by the EPR effect is still limited. Therefore, additional efforts such as surface modifications by targeting ligands have been paid to enhance the active targeting of CCM-loaded nanocarriers. Even so, these synthetic nanovectors fail to accurately mimic the natural complex interfaces; thus, they may be treated as foreign bodies, finally inducing an immune response and numerous side effects [19].

Cell membrane bioinspired nanocarriers have emerged as a novel platform and exhibit many advantages over synthetic delivery systems in terms of characteristics such as the prolonged circulation, excellent biocompatibility, low immunogenicity, and sophisticated functions [20]. In particular, platelet membrane (PM) camouflaged nanoparticles commendably improve the in vivo kinetic characteristics of cargoes, which is beneficial for subsequent therapies [19,21]. Additionally, PM biomimetic nanoparticles have been widely used in active targeting cancer therapy [22,23,24]. Nevertheless, most of these studies have only focused on engineering PM bioinspired nanoparticles exquisitely with the purpose of better targeted treatment for a particular type of cancer [23,25,26]. Only limited efforts have been devoted to systematic exploration on the binding affinity of the PM cloaked nanoparticle toward different tumor cells and the related mechanism.

Herein, the CCM-encapsulated lipid vesicle (CCM@Lip) was camouflaged by PM to fabricate a CCM-encapsulated PM biomimetic lipid vesicle (CCM@PL) to enhance the pharmacokinetics, cellular uptake, and anticancer cytotoxicity of CCM (Figure 1). CD44 highly expressed tumor cells including Huh-7 cells [27], SK-OV-3 cells [28], and MDA-MB-231 cells [29] were used to evaluate the tumor affinity of the PM biomimetic lipid vesicle (PL). Following intravenous injection, CCM@PL possessed long-term circulation and favorable bioavailability. Additionally, PL exhibited high affinity for tumor cells, which was likely attributed to the interaction between tumoral CD44 and platelet P-selectin. Thus, the PL displayed improved cellular uptake by the tumor cells. Furthermore, CCM@PL showed enhanced anticancer activity in Huh-7, SK-OV-3, and MDA-MB-231 cells. Altogether, this PM biomimicry strategy may provide a promising approach for cancer targeting therapy and natural components delivery.

## 2. Materials and Methods

### 2.1. Materials

Cholesterol was purchased from Sigma–Aldrich Chemical Co. (St. Louis, MO, USA). CCM was obtained from Yuanye Biotechnology Co., Ltd. (Shanghai, China). Hyaluronic acid (HA), D-α-tocopheryl polyethylene glycol 1000 succinate (TPGS), bicinchoninic acid (BCA) protein assay kits, and phospholipids were purchased from Meilun Biotechnology Co., Ltd. (Dalian, Liaoning, China). Phosphate buffered saline (PBS) was purchased from Solarbio Science & Technology Co., Ltd. (Beijing, China). Dulbecco’s modified Eagle’s medium (DMEM) and Fetal bovine serum (FBS) were purchased from GIBCO Invitrogen Corp (Carlsbad, CA, USA). 1,1′-Dioctadecyl-3,3,3′,3′-tetramethylindodicarbocyanine, 4-chlorobenzenesulfonate salt (DiD) and 1,1′-dioctadecyl-3,3,3′,3′-tetramethylindotricarbocyanine iodide (DiR) were purchased from Lablead Biotech Co., Ltd. (Beijing, China). Hoechst 33342 was purchased from Cell Signaling Technology Inc. (Danvers, MA, USA). Mouse anti-P-selectin antibody and rabbit anti-CD47 antibody were purchased from Proteintech Group, Inc. (Wuhan, Hubei, China).

Female Sprague Dawley rats (200 ± 10 g) were supplied by the Laboratory Animal Center at Southwest Medical University (China). All animals were housed under specific pathogen-free (SPF) conditions.

Hepatocellular carcinoma cell line (Huh-7), human ovarian cancer cell line (SK-OV-3), and human breast cancer cell line (MDA-MB-231) were obtained from the American Type Culture Collection (ATCC). Human umbilical vein endothelial cells (HUVECs) were a generous gift from Laboratory for Cardiovascular Pharmacology of Department of Pharmacology, School of Pharmacy, Southwest Medical University, China. All cells were cultured in DMEM supplemented with 10% FBS at 37 °C and under a 5% CO_2_ atmosphere.

### 2.2. Preparation of CCM@PL

CCM@Lip was prepared using a thin film dispersion method with some modifications [9,15]. In brief, the CCM, cholesterol, phospholipids, and TPGS were mixed in a round-bottom flask containing 30 mL of chloroform, then a thin film was obtained using rotary vacuum evaporation. Then, the resulting film was hydrated with 10 mL of PBS for 120 min followed by sonication for 5 min to fabricate CCM@Lip. The PMs were prepared through the method described in a previous report with slight modification [19]. Briefly, platelets from the rat whole blood were isolated by differential centrifugation. The platelets were frozen (−80 °C) and thawed, and this process was repeated three times. Then, the membranes were acquired by washing with PBS containing protease inhibitors and sonication for 5 min. To prepare CCM@PL, the obtained PM was mixed with CCM@Lip followed by sonication for 5 min [25]. The size distribution and zeta potential of CCM@PL were assessed via dynamic light scattering (Zeta-Sizer Nano-ZS, Malvern, Worcestershire, UK). The micromorphological feature of CCM@PL was investigated by a transmission electron microscopy (HT7700, Hitachi, Tokyo, Japan).

### 2.3. Protein Environment

The membrane protein composition of CCM@PL was characterized using sodium dodecyl sulfate–polyacrylamide gel electrophoresis (SDS–PAGE) and Western blot analysis. Briefly, the PM and CCM@PL were normalized to equivalent overall protein concentrations using a BCA protein assay kit. Proteins were then separated using 10% SDS–PAGE gel, followed by staining with Coomassie Blue and imaging. Western blotting was performed to detect the specific protein markers (CD47 and P-selectin). Markers of platelet membrane proteins were identified using antibodies against CD47 and P-selectin.

### 2.4. Encapsulation Efficiency

CCM@PL was subjected to centrifuge at 20,000× *g* for 40 min to separate the free CCM from the encapsulated drug. The amount of free CCM in the supernatant was determined spectrophotometrically at 426 nm. The mass of encapsulated CCM was calculated by subtracting the free CCM from the total input. The encapsulation efficiency was measured according to Equation (1):(1)$$\mathrm{Encapsulation}\;\mathrm{efficiency}\;(\%)\;=\;\frac{\mathrm{Mass}\;\mathrm{of}\;\mathrm{drug}\;\mathrm{encapsulated}\;\mathrm{in}\;\mathrm{PL}}{\mathrm{Total}\;\mathrm{mass}\;\mathrm{of}\;\mathrm{drug}}\times \;100\%$$

### 2.5. In Vitro Release

The in vitro release of CCM from CCM@PL was conducted using the dialysis method with some modifications [9,16]. Dialysis bags were activated and immersed in release media (pH 7.4) overnight before use [17]. One mL of CCM@PL was taken in the dialysis bag (molecular weight cutoff, 8000 Da) and the bags were immersed into 100 mL of PBS release media (pH 7.4) having 0.5% sodium dodecyl sulfate and 20% ethanol. The experiment was carried out at 37 °C in dark with a shaking speed of 100 rpm. At specified time intervals (1, 2, 4, 6, 8, 10, 12, 24, 48, 72, 96 h), 1 mL of release media was withdrawn and an equal volume of fresh release medium was added to maintain constant volume and sink condition. The concentration of CCM was measured spectrophotometrically at 426 nm [30]. The release experiment was performed in triplicate.

### 2.6. Pharmacokinetics

#### 2.6.1. Animal Experiment

Female Sprague Dawley rats were randomly divided into two groups (four rats per group) intravenously administered with CCM or CCM@PL at the same CCM dose (8 mg/kg). At the predetermined time points (0.17, 0.5, 1, 2, 3, 4, 6, 8 h), venous blood samples were collected into heparinized tubes and then centrifuged at 5000 rpm for 5 min. The supernatants were collected to obtain plasma samples, and these samples were detected by a high-performance liquid chromatography (HPLC) method with slight modifications [31]. The primary pharmacokinetic parameters including area under the concentration-time curve (*AUC*), clearance (*Cl*), and mean residence time (*MRT*) were obtained by a DAS software 3.0 (Mathematical Pharmacology Professional Committee of China, Shanghai, China).

#### 2.6.2. Sample Preparation

A mixture of ethyl acetate and methanol (9: 1, *v*/*v*) were added to the plasma samples, following by vortex for 5 min. After centrifugation at 12,000 rpm for 10 min, the supernatant was obtained and then subjected to evaporate for dryness at 37 °C under nitrogen. The residue was redissolved and centrifuged at 12,000 rpm for 10 min. Then, the supernatant was transferred to a glass tube and a 20 μL volume was injected into the HPLC system.

#### 2.6.3. HPLC Analysis

CCM was assayed using a HPLC system (Agilent-1260, Santa Clara, CA, USA). This analysis was performed at 426 nm with a Kromasil 100-5-C18, 4.6 × 250 mm column maintained at 30 °C. The mobile phase was composed of 5% acetic acid and acetonitrile (50:50, *v*/*v*) at a flow rate of 1 mL/min.

### 2.7. In Vivo Distribution

Free DiR or DiR-encapsulated PM biomimetic lipid vesicle (DiR@PL) was intravenously given to rats at 0.2 mg/kg of DiR (*n* = 4). For fluorescence imaging visualization, blood samples were repeatedly collected before treatment and at 1, 2, 3, 4, 6, and 8 h post injection. Then, the blood samples were subjected to centrifuge at 5000 rpm for 5 min to obtain plasma. These plasma samples were analyzed using an in vivo imaging system (FXPRO, Bruker, Billerica, MA, USA) to detect the DiR signal. The fluorescence intensities from plasma samples were also measured by a microplate reader (VLBL00D0, Thermo Fisher, Waltham, MA, USA). At the end of the experiment, the rats were autopsied and the major organs were collected. The fluorescence signals of each organ were recorded using the Bruker imaging system for quantification.

### 2.8. Cytotoxicity

The cytotoxicity of formulations against CD44-expressing cells including Huh-7 cells, SK-OV-3 cells, and MDA-MB-231 cells was determined using an MTT assay. Cells (5 × 10^3^ cells/well) were prepared, and then CCM, blank PL, CCM@Lip, and CCM@PL (CCM concentration: 20 µM) were added to the cells. At 24 h post incubation, MTT solution (0.5 mg/mL) was added and the plate was incubated for 4 h. The formazan product was solubilized with DMSO (150 µL). Then, the absorbance was measured with a microplate reader at 570 nm.

### 2.9. Cytocompatibility

To evaluate the in vitro cytocompatibility of blank PL, HUVECs were cultured in 96-well plates at a density of 5 × 10^3^ cells/well, following incubation with PL (0, 50, 100, 200, or 400 μg/mL) for 24 h. After treatment, cells were determined by MTT method as described above.

### 2.10. Cellular Uptake

For cellular uptake evaluation, 200 μL (5 × 10^4^ cells/well) of Huh-7, SK-OV-3, or MDA-MB-231 cells were seeded onto confocal dishes and cultured for 12 h. The cells were then treated with the DiD-encapsulated PM biomimetic lipid vesicle (DiD@PL) or DiD-encapsulated lipid vesicle (DiD@Lip) for 3 h. For confocal imaging, after removing the culture media, cells were fixed with 4% paraformaldehyde for 20 min, stained with 0.5 mL of Hoechst 33342 for 3 min, and rinsed with PBS. For the blocking experiments, Huh-7, SK-OV-3, or MDA-MB-231 cells were preincubated with free HA (10 mg/mL) for 3 h. After removing the free HA, the cells were incubated with DiD@PL for another 3 h. The cells were then treated and imaged as described above.

### 2.11. Statistical Analysis

Data were presented as the mean ± standard deviation (SD). Comparisons were performed using one-way ANOVA, and Turkey’s post-hoc test. * *p* < 0.05, ** *p* < 0.01, and *** *p* < 0.001.

## 3. Results and Discussion

### 3.1. Morphology, Physicochemical Properties, and In Vitro Release

When examined by transmission electron microscopy, CCM@PL exhibited a sphere-like structure (Figure 1A), consistent with previous reports [19,32]. The mean size and zeta potential of CCM@PL were 163.2 nm and −31.8 mV, respectively. CCM@PL showed high encapsulation efficiency with a value of 93.62%.

SDS-PAGE results showed protein profiles of PM were well-preserved in the CCM@PL (Figure 1B). Additionally, the Western blotting was performed to determine the platelet-specific proteins on the CCM@PL. As shown in Figure 1C, cancer-targeted protein P-selectin and “self-recognized” protein CD47 were present on CCM@PL. These results suggest that PM proteins had been transferred to the resultant nanocarrier.

The release of CCM from CCM@PL was profiled over time at a physiological pH of 7.4 (Figure 1D). A rapid burst was observed in the first 24 h, with slow drug release plateauing near 50% over the next 48 h. Over the course of 4 days, approximately 50% of the loaded CCM was retained, and this result was similar to the previous studies, indicating that the nanovesicle was quite stable and could be used for achieving prolonged delivery of the payloads [12,33,34].

### 3.2. In Vivo Pharmacokinetics and Biodistribution

The plasma drug concentration-time profiles (Figure 2A) and the main pharmacokinetic parameters (Figure 2B–D) suggested that the pharmacokinetic behavior of CCM had been favorably modified by loading it inside CCM@PL after intravenous injection. Compared to free CCM, the *AUC* value of CCM@PL was 3.08-fold that of CCM. The *Cl* value of free CCM was 2.73-fold that of CCM@PL. CCM@PL had extended blood retention times compared with free CCM, with a *MRT* value of 2.89 h, which was 3.04 times that of free CCM. This finding confirmed that CCM@PL could provide long blood circulation. This was probably because the CD47 protein on the PM could help biomimetic nanoparticle escape the uptake by macrophage to prolong the circulation time [35,36].

To further evaluate the in vivo behavior of PL, we next investigated the plasma pharmacokinetics of DiR@PL. The fluorescence intensities of plasma samples were measured by a microplate reader, and the result showed that the fluorescence intensity from the DiR@PL group was higher than that from the DiR group at 2, 3, 4, 6, and 8 h after administration, indicating the prolonged circulation of DiR@PL (Figure 3A). In addition, the result obtained by the imaging system further confirmed the in vivo long retention of DiR@PL (Figure 3B,C). As shown in (Figure 3D,E), the fluorescence signals of free DiR were lower in the tissues than those in DiR@PL due to the rapid metabolism of bulk drugs. The longer-term maintained concentration of cargoes might increase the therapeutic efficiency.

### 3.3. Enhanced Cytotoxicity and Cellular Uptake

The in vitro cytotoxicity of CCM@PL against Huh-7, SK-OV-3, or MDA-MB-231 cells was evaluated using an MTT assay. Blank PL did not show cytotoxicity. However, CCM@PL significantly enhanced cytotoxicity compared with free CCM and CCM@Lip (Figure 4).

Subsequently, the cellular uptake was evaluated. Higher fluorescence intensity was found in the cancer cells treated with PL (Figure 5), suggesting that PL was more efficiently taken up by the tumor cells. To further evaluate the endocytosis mechanism of cellular uptake, a blocking assay was performed. Huh-7, SK-OV-3, and MDA-MB-231 cells were pretreated with free HA, which is a high-affinity ligand of the CD44 receptor [37]. The fluorescence signal of PL in free HA preincubated cancer cells was reduced (Figure 5). Previous studies have shown that there is a specific and strong affinity between the CD44 receptor and P-selectin [23,29]. These results indicated that the biomimetic membrane could enhance internalization by cancerous cells, presumably attributed to the binding of P-selectin on the membrane to CD44 receptors of tumor cells. The receptor-mediated uptake of CCM@PL is a useful feature in inducing cancer cell selective cytotoxicity. Collectively, based on the higher cellular uptake, these results showed that PL formulation conferred high cytotoxicity to the CD44 receptor overexpressing cancer cells.

### 3.4. Cytocompatibility

To further evaluate the potential of PL in pharmaceutical applications, the in vitro cytocompatibility of blank PL on HUVECs was examined. There was no cytotoxicity after blank PL treatment with HUVECs even when the concentration of vesicles reached 400 μg/mL (Figure 6), indicating that such pure vesicles could be regarded as good cytocompatibility.

## 4. Conclusions

The developed formulation CCM@PL showed prolonged circulation time and improved bioavailability. Moreover, PL exhibited good targeting capacity toward Huh-7, SK-OV-3, and MDA-MB-231 cells. This might be attributed to the biomolecular binding between platelet P-selectin and cancerous CD44. In addition, CCM@PL exhibited remarkable cytotoxic effects on cancer cells. Therefore, CCM@PL has been speculated to be an encouraging application for tumor targeted therapy. Future studies are planned to confirm the in vivo targeting capacity of CCM@PL as well as in vivo anticancer effect.

Thus far, the anticancer effect of the emerging biomimetic nanomedicine has not been fully demonstrated in humans. In our work, CCM@PL was prepared using a platelet membrane of rat for preliminary lab research, which is not suitable for human use. In future, we will use a human platelet membrane to improve this bioinspired formulation for further research.

## Data Availability

Data are contained within the article or from the authors upon reasonable request.

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
