# Peer review of "Bioinspired Platelet-like Nanovector for Enhancing Cancer Therapy via P-Selectin Targeting"

_pharmaceutics, 2022, doi:10.3390/pharmaceutics14122614_

Round 1

Reviewer 1 Report

The manuscript "Bioinspired Platelet-like Nanovector for Enhancing Cancer Therapy via P-selectin Targeting" deals with development of platelet-like nanovesicle of curcumin (CCM) for enhanced cancer therapy via p-selectin targeting. The prepared nanovesicles were characterized physicochemically and evaluated for anticancer potential and pharmacokinetic studies. The paper is well organized and well written, covering the most significant topics in the field. However, there are several changes authors should make before the manuscript can be accepted for publication:

Abstract: Quantitative results are completely missing in the abstract. The authors are advised to include some quantitative results in order to enhance the readability and understanding of the manuscript.

Avoid abbreviations before giving their explanation.

The authors should suggest, based on this article, an area of ​​research or space that still needs to be worked on.

Introduction: The authors are advised to include the existing nanosystems of CCM and include the advantages of present drug delivery system over the existing drug delivery systems of CCM.

Section 2.6. In vitro release, authors are advised to include more details about this section.

Section 2.7. Pharmacokinetics, this section is too poor. Please include the detailed procedure. Also include the details about HPLC method used for the analysis of CCM in plasma. HPLC method was reported one or laboratory developed. Please include in details. What was the internal standard used for the recovery of CCM?

Figures should be given in better resolution. 

Some important literature about the proposed topic are missing. Authors are advised to add these literature to make this manuscript more useful to the readers:

J. Liposome Res. 24: 53-58 (2014); J. Pharm. Sci. 106: 3050-3065 (2017); Curr. Drug Deliv. 17: 898-910 (2020); Curr. Drug Deliv. 17: 826-844 (2020); Pharmaceutics 13: E368 (2021); Molecules 27: E733 (2022); Molecules 27: E6608 (2022); J. Mol. Liq. 348: E118008 (2022); J. Mol. Liq. 350: E118189 (2022).

Authors should briefly comment on the disadvantages of the studied systems. One whole paragraph should be added to let readers get better insight into the subject.

Conclusion: The conclusion should be concise and to the point indicating the application of the work.

Author Response

Nov. 20, 2022

 Dear Expert Reviewer,

Thank you very much for the prompt review process and excellent comments. We greatly appreciate the time and efforts which you have spent on it. We are submitting the revised manuscript entitled “Bioinspired Platelet-like Nanovector for Enhancing Cancer Therapy via P-selectin Targeting” (ID: pharmaceutics-1962608) to Pharmaceutics.

We have carefully considered your comments and suggestions, and addressed each of the concerns in response to the comments (see point by point response). We have revised the manuscripts based on your comments and carefully checked throughout the manuscript and corrected the language errors. Our point-by-point responses to the comments (in blue) are shown below (in red).

Comment 1. Abstract: Quantitative results are completely missing in the abstract. The authors are advised to include some quantitative results in order to enhance the readability and understanding of the manuscript.

Response 1: Thank you very much for the valuable advice. We are very sorry for the missing of quantitative results in the abstract. And we have added quantitative results in the abstract section (page 1, lines 28-31) as follows:

In this work, CCM-encapsulated platelet membrane biomimetic lipid vesicle (CCM@PL) with size of 163.2 nm, zeta potential of -31.8 mV and encapsulation efficiency of 93.62% was developed. The values of area under the concentration-time curve (AUC) and mean residence time (MRT) for CCM@PL were 3.08 times and 3.04 times those of CCM, respectively.

Comment 2. Avoid abbreviations before giving their explanation.

Response 2: Thanks for your careful review. We are very sorry for used abbreviations before giving their explanations. And we have given the explanations before using abbreviations in the revised manuscript. The revised abbreviations are as follows:

Table R1. Examples of abbreviations corrections

Location in the revised manuscript

Original text

Correction

page 1, lines 28-29

CCM@PL

CCM-encapsulated platelet membrane biomimetic lipid vesicle (CCM@PL)

page 2, lines 75-76

PL

PM biomimetic lipid vesicle (PL)

page 4, line 144

MWCO

molecular weight cutoff

page 5, line 158

HPLC

high-performance liquid chromatography (HPLC)

page 5, line 176

DiR@PL

DiR-encapsulated PM biomimetic lipid vesicle (DiR@PL)

page 5, line 202

DiD@PL

DiD-encapsulated PM biomimetic lipid vesicle (DiD@PL)

page 5, lines 202-203

DiD@Lip

DiD-encapsulated lipid vesicle (DiD@Lip)

page 7, line 246

SD rats

Sprague Dawley rats

page 8, line 259

SD rats

Sprague Dawley rats

Comment 3. The authors should suggest, based on this article, an area of research or space that still needs to be worked on.

Response 3: Thanks a lot for the constructive and careful suggestion. Based on the results in our work, CCM@PL is an encouraging approach applied in the targeted cancer treatment. Future studies are planned to investigate the in vivo targeting capacity of CCM@PL as well as in vivo anticancer effect. Thus far, the anticancer effect of the emerging biomimetic nanomedicine has not been fully demonstrated in humans. In our work, CCM@PL was prepared using platelet membrane of rat for preliminary lab research, which is not suitable for human use. In the future, we will use human platelet membrane to improve this bioinspired formulation for further research. Meanwhile, we have added the above discussions in the conclusion section (page 10, lines 304-311).

Comment 4. Introduction: The authors are advised to include the existing nanosystems of CCM and include the advantages of present drug delivery system over the existing drug delivery systems of CCM.

Response 4: We gratefully appreciate your valuable suggestion. We have added the existing nanosystems of CCM, and advantages of the developed drug delivery system over the existing drug delivery systems of CCM were discussed in the introduction section (page 2, lines 49-66). The revised content is as follow:

Many studies have been focused on the development of CCM-loaded nanocarriers, like micelles, polymeric nanoparticles, silica nanoparticles, and lipid vesicles [7,9]. Particularly, nano lipid vesicle mainly composed with phospholipid and cholesterol are used as an efficient delivery of CCM to enhance the effect of CCM [8,18]. Besides, nano-vectors can enhance CCM tumor accumulation via the enhanced permeability and retention (EPR) effect. Nevertheless, this passive tumor targeting caused by EPR effect is still limited. Therefore, additional efforts such as surface modifications by targeting ligands have been paid to enhance the active targeting of CCM-loaded nanocarriers. Even so, these synthetic nano-vectors fail to accurately mimic the natural complex interfaces, thus they may be treated as foreign bodies, finally inducing an immune response and numerous side effects [19]. Cell membrane bioinspired nanocarriers have emerged as a novel platform and exhibit many advantages over synthetic delivery systems in terms of characteristics such as the prolonged circulation, excellent biocompatibility, low immunogenicity, and sophisticated functions [20]. In particular, platelet membrane (PM) camouflaged nanoparticles commendably improve the in vivo kinetic characteristics of cargoes, which is beneficial for subsequent therapies [19,21]. Additionally, PM biomimetic nanoparticles have been widely used in active targeting cancer therapy [22-24].

References

  1. Shoaib, A.; Azmi, L.; Pal, S.; Alqahtani, S.S.; Rahamathulla, M.; Hani, U.; Alshehri, S.; Ghoneim, M.M.; Shakeel, F. Integrating nanotechnology with naturally occurring phytochemicalsin neuropathy induced by diabetes. Journal of Molecular Liquids 2022, 350, 118189.
  2. Wang, Y.; Ding, R.; Zhang, Z.; Zhong, C.; Wang, J.; Wang, M. Curcumin-loaded liposomes with the hepatic and lysosomal dual-targeted effects for therapy of hepatocellular carcinoma. Int J Pharm 2021, 602, 120628.
  3. Xie, X.; Li, Y.; Zhao, D.; Fang, C.; He, D.; Yang, Q.; Yang, L.; Chen, R.; Tan, Q.; Zhang, J. Oral administration of natural polyphenol-loaded natural polysaccharide-cloaked lipidic nanocarriers to improve efficacy against small-cell lung cancer. Nanomedicine 2020, 29, 102261.
  4. Alanazi, F.K.; Lu, D.R.; Shakeel, F.; Haq, N. Density gradient separation of carborane-containing liposome from low density lipoprotein and detection by inductively coupled plasma spectrometry. Journal of liposome research 2014, 24, 53-58.
  5. Liu, G.; Zhao, X.; Zhang, Y.; Xu, J.; Xu, J.; Li, Y.; Min, H.; Shi, J.; Zhao, Y.; Wei, J., et al. Engineering biomimetic platesomes for pH-responsive drug delivery and enhanced antitumor activity. Adv Mater 2019, 31, e1900795.
  6. Javed, S.; Alshehri, S.; Shoaib, A.; Ahsan, W.; Sultan, M.H.; Alqahtani, S.S.; Kazi, M.; Shakeel, F. Chronicles of nanoerythrosomes: An erythrocyte-based biomimetic smart drug delivery system as a therapeutic and diagnostic tool in cancer therapy. Pharmaceutics 2021, 13.
  7. Pei, W.; Huang, B.; Chen, S.; Wang, L.; Xu, Y.; Niu, C. Platelet-mimicking drug delivery nanoparticles for enhanced chemo-photothermal therapy of breast cancer. Int J Nanomedicine 2020, 15, 10151-10167.
  8. Hu, Q.; Sun, W.; Qian, C.; Wang, C.; Bomba, H.N.; Gu, Z. Anticancer platelet-mimicking nanovehicles. Adv Mater 2015, 27, 7043-7050.
  9. Wang, H.; Wu, J.; Williams, G.R.; Fan, Q.; Niu, S.; Wu, J.; Xie, X.; Zhu, L.M. Platelet-membrane-biomimetic nanoparticles for targeted antitumor drug delivery. J Nanobiotechnology 2019, 17, 60.
  10. Liao, Y.; Zhang, Y.; Blum, N.T.; Lin, J.; Huang, P. Biomimetic hybrid membrane-based nanoplatforms: Synthesis, properties and biomedical applications. Nanoscale horizons 2020, 5, 1293-1302.

Comment 5. Section 2.6. In vitro release, authors are advised to include more details about this section.

Response 5: Thanks a lot for the constructive and careful suggestion. We are very sorry for this poor section, and we have added more details about the release section (page 4, lines140-150) as follows:

2.5 In vitro release

The in vitro release of CCM from CCM@PL was conducted using the dialysis method with some modifications [9,16]. Dialysis bags were activated and immersed in release media (pH 7.4) overnight before use [17]. 1 mL of CCM@PL was taken in the dialysis bag (molecular weight cutoff, 8,000 Da) and the bags were immersed into 100 mL of PBS release media (pH 7.4) having 0.5% sodium dodecyl sulfate and 20% ethanol. The experiment was carried out at 37 °C in dark with a shaking speed of 100 rpm. At specified time intervals (1, 2, 4, 6, 8, 10, 12, 24, 48, 72, 96 h), 1 mL of release media was withdrawn and an equal volume of fresh release medium at 37 °C was added to maintain constant volume and sink condition. The concentration of CCM was measured spectrophotometrically at 426 nm [30]. The release experiment was performed in triplicate.

References

  1. Xie, X.; Li, Y.; Zhao, D.; Fang, C.; He, D.; Yang, Q.; Yang, L.; Chen, R.; Tan, Q.; Zhang, J. Oral administration of natural polyphenol-loaded natural polysaccharide-cloaked lipidic nanocarriers to improve efficacy against small-cell lung cancer. Nanomedicine 2020, 29, 102261.
  2. Muheem, A.; Shakeel, F.; Warsi, M.H.; Jain, G.K.; Ahmad, F.J. A combinatorial statistical design approach to optimize the nanostructured cubosomal carrier system for oral delivery of ubidecarenone for management of doxorubicin-induced cardiotoxicity: In vitro-in vivo investigations. Journal of pharmaceutical sciences 2017, 106, 3050-3065.
  3. Al-Joufi, F.A.; Salem-Bekhit, M.M.; Taha, E.I.; Ibrahim, M.A.; Muharram, M.M.; Alshehri, S.; Ghoneim, M.M.; Shakeel, F. Enhancing ocular bioavailability of ciprofloxacin using colloidal lipid-based carrier for the management of post-surgical infection. Molecules (Basel, Switzerland) 2022, 27.
  4. Zhu, J.; Wang, Y.; Yang, P.; Liu, Q.; Hu, J.; Yang, W.; Liu, P.; He, F.; Bai, Y.; Gai, S., et al. Gpc3-targeted and curcumin-loaded phospholipid microbubbles for sono-photodynamic therapy in liver cancer cells. Colloids Surf B Biointerfaces 2021, 197, 111358.

Comment 6. Section 2.7. Pharmacokinetics, this section is too poor. Please include the detailed procedure. Also include the details about HPLC method used for the analysis of CCM in plasma. HPLC method was reported one or laboratory developed. Please include in details. What was the internal standard used for the recovery of CCM?

Response 6: Thank you so much for your scientific review. We are very sorry for this poor section. And we have added the detailed procedure about pharmacokinetics including details about HPLC method used for the analysis of CCM in plasma (pages 4-5, lines 151-174).

  • The detailed procedure for pharmacokinetic is as follow:

Female Sprague Dawley rats were randomly divided into two groups (four rats per group) to intravenously administered with CCM or CCM@PL at the same CCM dose (8 mg/kg). At the predetermined time points (0.17, 0.5, 1, 2, 3, 4, 6, 8 h), venous blood samples were collected into heparinized tubes and then centrifuged at 5,000 rpm for 5 min. The supernatants were collected to obtain plasma samples, and these samples were detected by a high-performance liquid chromatography (HPLC) method with slight modifications (Pi. et al. Drug Deliv 2022). The primary pharmacokinetic parameters including area under the concentration-time curve (AUC), clearance (Cl), and mean residence time (MRT) were obtained by a DAS software (Mathematical Pharmacology Professional Committee of China, Shanghai, China).

For sample preparation, a mixture of ethyl acetate and methanol (9: 1, v/v) were added to the plasma samples, following by vortex for 5 min. After centrifugation at 12,000 rpm for 10 min, the supernatant was obtained and then subjected to evaporate for dryness at 37°C under nitrogen. The residue was redissolved and centrifuged at 12,000 rpm for 10 min. And then the supernatant was transferred to glass tube and a 20 μL volume was injected into the HPLC system.

  • The HPLC method used for the analysis of CCM in plasma is as follow:

The plasma samples were detected by a HPLC method with slight modifications (Pi. et al. Drug Deliv 2022). CCM was assayed using a HPLC system (Agilent-1260, CA, USA). This analysis was performed at 426 nm with a Kromasil 100-5-C18, 4.6×250 mm column maintained at 30°C. The mobile phase was composed of 5% acetic acid and acetonitrile (50:50, v/v) at a flow rate of 1 mL/min.

The specificity was assessed by analyzing the samples of blank plasma, blank plasma spiked with CUR, and the plasma obtained from rats after intravenous injection of CCM@PL. The typical chromatograms of blank rat plasma, spiked rat plasma, and test rat plasma are shown in Figure R1. No other interfering peaks were detected.

Figure R1 please find it in the  attachment word file.

Figure R1 The typical HPLC chromatograms of (A) blank plasma, (B) blank plasma spiked with CCM, and (C) sample obtained from rats administrated with CCM@PL.

Linearity of the analyte was evaluated by analyzing a series of standard concentrations (10, 20, 40, 80, 160, 320, 640, 1280 and 2560 ng/mL) in rat plasma. The nine-point calibration curve for the plasma samples was obtained by plotting the peak area of CCM versus the nominal concentration using the least-squares method. CCM plasma concentration could be detected by this HPLC method, within a good linearity in the range of 10-2560 ng/mL (y = 0.1111 x - 0.4227, r = 0.9997, n = 3).

  • What was the internal standard used for the recovery of CCM?

Thanks for your good question. It is rigorous for using the internal standard to evaluate the recovery of analyte. In this work, the recovery of the extracted samples was calculated by analyzing extracted samples of the low, medium and high concentrations (80, 640 and 1280 ng/mL) and then comparing the peak area ratio of these samples with those of equivalent unextracted CCM samples. The method recoveries of low, middle, high concentrations were 95.83%, 104.88%, 101.98%, respectively, which in the range of 85%~115%. Therefore, this method with high recovery was suitable for detection of CCM. In some pharmacokinetic studies, the internal standards were also not added for investigating the analyte extraction, but the analyte recovery was high and acceptable (Pi. et al. Drug Deliv 2022, Guo. et al. Expert opinion on drug delivery 2020, Nadaf. et al. Acta chimica Slovenica 2020, Shoba. et al. Planta medica 1998). Thank you so much for your scientific review, we will add internal standards to evaluate analyte recovery in future research work.

References

Pi, C.; Zhao, W.; Zeng, M.; Yuan, J.; Shen, H.; Li, K.; Su, Z.; Liu, Z.; Wen, J.; Song, X., et al. Anti-lung cancer effect of paclitaxel solid lipid nanoparticles delivery system with curcumin as co-loading partner in vitro and in vivo. Drug Deliv 2022, 29, 1878-1891.  (Pi. et al. Drug Deliv 2022)

Guo, P.; Pi, C.; Zhao, S.; Fu, S.; Yang, H.; Zheng, X.; Zhang, X.; Zhao, L.; Wei, Y. Oral co-delivery nanoemulsion of 5-fluorouracil and curcumin for synergistic effects against liver cancer. Expert opinion on drug delivery 2020, 17, 1473-1484.  (Guo. et al. Expert opinion on drug delivery 2020)

Nadaf, S.; Killedar, S. Development and validation of rp–hplc method for estimation of curcumin from nanocochleates and its application in in–vivo pharmacokinetic study. Acta chimica Slovenica 2020, 67, 1100-1110.  (Nadaf. et al. Acta chimica Slovenica 2020)

Shoba, G.; Joy, D.; Joseph, T.; Majeed, M.; Rajendran, R.; Srinivas, P.S. Influence of piperine on the pharmacokinetics of curcumin in animals and human volunteers. Planta medica 1998, 64, 353-356.  (Shoba. et al. Planta medica 1998)

Comment 7. Figures should be given in better resolution.

Response 7: Thanks for your careful review. We are very sorry that there are low-resolution figures. In the revised manuscript, we changed the size and improved the resolution of the images on (page 3, Scheme 1), (page 6, Figure 1), (page 7, Figure 2), (page 8, Figure 3), (page 9, Figure 4), (page 9, Figure 5), and (page 10, Figure 6). Meanwhile, we re-uploaded the revised figures in the submission system, and corrected the misplaced order images in Figure 5.

Comment 8. Some important literature about the proposed topic are missing. Authors are advised to add these literature to make this manuscript more useful to the readers:

  1. Liposome Res. 24: 53-58 (2014); J. Pharm. Sci. 106: 3050-3065 (2017); Curr. Drug Deliv. 17: 898-910 (2020); Curr. Drug Deliv. 17: 826-844 (2020); Pharmaceutics 13: E368 (2021); Molecules 27: E733 (2022); Molecules 27: E6608 (2022); J. Mol. Liq. 348: E118008 (2022); J. Mol. Liq. 350: E118189 (2022)

Response 8: Thanks for your kindly attention and consideration. According to your comments, we have added all these literatures in the introduction section (pages 1-2, lines 39-82) and some related literatures have also been added in the method section (page 4, lines 112-113 and lines 141-143) to make this manuscript more useful to the readers. The introduction in our manuscript was revised as follows:

Despite the advanced science and technology, cancer is still a disease burden worldwide [1,2]. Conventional methods for cancer treatment include surgery, chemotherapy, and radiation therapy. Traditional chemotherapy has severe problems such as poor specificity and undesirable toxicity. Nowadays, various natural products are important resources against tumor growth [3-6]. Curcumin (CCM) is a natural phenolic compound extracted from Curcuma longa, possessing favorable anticancer effect with non-toxic [7-10]. However, the therapeutic potentials of CCM are universally limited by the low bioavailability, poor tumor affinity, and restricted cellular uptake [11,12] Nanodrug delivery systems have attracted tremendous attentions in improving the pharmacokinetics and tumor targeting of chemotherapeutics including phytochemicals [13-17]. Many studies have been focused on the development of CCM-loaded nanocarriers, like micelles, polymeric nanoparticles, silica nanoparticles, and lipid vesicles [7,9]. Particularly, nano lipid vesicle mainly composed with phospholipid and cholesterol are used as an efficient delivery of CCM to enhance the effect of CCM [8,18]. Besides, nano-vectors can enhance CCM tumor accumulation via the enhanced permeability and retention (EPR) effect. Nevertheless, this passive tumor targeting caused by EPR effect is still limited. Therefore, additional efforts such as surface modifications by targeting ligands have been paid to enhance the active targeting of CCM-loaded nanocarriers. Even so, these synthetic nanovectors fail to accurately mimic the natural complex interfaces, thus they may be treated as foreign bodies, finally inducing an immune response and numerous side effects [19].

Cell membrane bioinspired nanocarriers have emerged as a novel platform and exhibit many advantages over synthetic delivery systems in terms of characteristics such as the prolonged circulation, excellent biocompatibility, low immunogenicity, and sophisticated functions [20]. In particular, platelet membrane (PM) camouflaged nanoparticles commendably improve the in vivo kinetic characteristics of cargoes, which is beneficial for subsequent therapies [19,21]. Additionally, PM biomimetic nanoparticles have been widely used in active targeting cancer therapy [22-24]. Nevertheless, most of these studies have only focused on engineering PM bioinspired nanoparticles exquisitely with the purpose of better targeted treatment for a particular type of cancer [23,25,26]. Only limited efforts have been devoted to systematic exploration on the binding affinity of PM cloaked nanoparticle toward different tumor cells and the related mechanism.

Herein, CCM-encapsulated lipid vesicle (CCM@Lip) was camouflaged by PM to fabricate CCM-encapsulated PM biomimetic lipid vesicle (CCM@PL) for enhancing the pharmacokinetics, cellular uptake, and anticancer cytotoxicity of CCM. CD44 highly expressed tumor cells including Huh-7 cells [27], SK-OV-3 cells [28], and MDA-MB-231 cells [29] were used to evaluate the tumor affinity of PM biomimetic lipid vesicle (PL). Following intravenous injection, PL possessed long-term circulation and favorable bioavailability. Additionally, PL exhibited high affinity for tumor cells, which was likely attributed to the interaction between tumoral CD44 and platelet P-selectin. Thus, the PL displayed improved cellular uptake by the tumor cells. Furthermore, PL showed enhanced anticancer activity in Huh-7, SK-OV-3, and MDA-MB-231 cells. Altogether, this PM biomimicry strategy may provide a promising approach for cancer targeting therapy and natural components delivery.

References

  1. Mir, S.A.; Hamid, L.; Bader, G.N.; Shoaib, A.; Rahamathulla, M.; Alshahrani, M.Y.; Alam, P.; Shakeel, F. Role of nanotechnology in overcoming the multidrug resistance in cancer therapy: A review. Molecules (Basel, Switzerland) 2022, 27.
  2. Sung, H.; Ferlay, J.; Siegel, R.L.; Laversanne, M.; Soerjomataram, I.; Jemal, A.; Bray, F. Global cancer statistics 2020: Globocan estimates of incidence and mortality worldwide for 36 cancers in 185 countries. CA: a cancer journal for clinicians 2021, 71, 209-249.
  3. Zhao, J.; Li, Y.; He, D.; Hu, X.; Li, K.; Yang, Q.; Fang, C.; Zhong, C.; Yang, J.; Tan, Q., et al. Natural oral anticancer medication in small ethanol nanosomes coated with a natural alkaline polysaccharide. ACS Appl Mater Interfaces 2020, 12, 16159-16167.
  4. Wu, J.; Williams, G.R.; Niu, S.; Gao, F.; Tang, R.; Zhu, L.M. A multifunctional biodegradable nanocomposite for cancer theranostics. Adv Sci (Weinh) 2019, 6, 1802001.
  5. Hu, Y.; Wang, S.; Wu, X.; Zhang, J.; Chen, R.; Chen, M.; Wang, Y. Chinese herbal medicine-derived compounds for cancer therapy: A focus on hepatocellular carcinoma. J Ethnopharmacol 2013, 149, 601-612.
  6. Zhou, P.; Li, J.; Chen, Q.; Wang, L.; Yang, J.; Wu, A.; Jiang, N.; Liu, Y.; Chen, J.; Zou, W., et al. A comprehensive review of genus sanguisorba: Traditional uses, chemical constituents and medical applications. Front Pharmacol 2021, 12, 750165.
  7. Shoaib, A.; Azmi, L.; Pal, S.; Alqahtani, S.S.; Rahamathulla, M.; Hani, U.; Alshehri, S.; Ghoneim, M.M.; Shakeel, F. Integrating nanotechnology with naturally occurring phytochemicalsin neuropathy induced by diabetes. Journal of Molecular Liquids 2022, 350, 118189.
  8. Wang, Y.; Ding, R.; Zhang, Z.; Zhong, C.; Wang, J.; Wang, M. Curcumin-loaded liposomes with the hepatic and lysosomal dual-targeted effects for therapy of hepatocellular carcinoma. Int J Pharm 2021, 602, 120628.
  9. Xie, X.; Li, Y.; Zhao, D.; Fang, C.; He, D.; Yang, Q.; Yang, L.; Chen, R.; Tan, Q.; Zhang, J. Oral administration of natural polyphenol-loaded natural polysaccharide-cloaked lipidic nanocarriers to improve efficacy against small-cell lung cancer. Nanomedicine 2020, 29, 102261.
  10. Tima, S.; Okonogi, S.; Ampasavate, C.; Berkland, C.; Anuchapreeda, S. Flt3-specific curcumin micelles enhance activity of curcumin on flt3-itd overexpressing mv4-11 leukemic cells. Drug Dev Ind Pharm 2019, 45, 498-505.
  11. Salarbashi, D.; Tafaghodi, M.; Fathi, M.; Aboutorabzade, S.M.; Sabbagh, F. Development of curcumin-loaded prunus armeniaca gum nanoparticles: Synthesis, characterization, control release behavior, and evaluation of anticancer and antimicrobial properties. Food science & nutrition 2021, 9, 6109-6119.
  12. Hong, W.; Guo, F.; Yu, N.; Ying, S.; Lou, B.; Wu, J.; Gao, Y.; Ji, X.; Wang, H.; Li, A., et al. A novel folic acid receptor-targeted drug delivery system based on curcumin-loaded β-cyclodextrin nanoparticles for cancer treatment. Drug design, development and therapy 2021, 15, 2843-2855.
  13. Khan, S.; Mansoor, S.; Rafi, Z.; Kumari, B.; Shoaib, A.; Saeed, M.; Alshehri, S.; Ghoneim, M.M.; Rahamathulla, M.; Hani, U., et al. A review on nanotechnology: Properties, applications, and mechanistic insights of cellular uptake mechanisms. Journal of Molecular Liquids 2022, 348, 118008.
  14. Alanazi, S.A.; Alanazi, F.; Haq, N.; Shakeel, F.; Badran, M.M.; Harisa, G.I. Lipoproteins-nanocarriers as a promising approach for targeting liver cancer: Present status and application prospects. Current drug delivery 2020, 17, 826-844.
  15. Alanazi, S.A.; Harisa, G.I.; Badran, M.M.; Haq, N.; Radwan, A.A.; Kumar, A.; Shakeel, F.; Alanazi, F.K. Cholesterol-conjugate as a new strategy to improve the cytotoxic effect of 5-fluorouracil on liver cancer: Impact of liposomal composition. Current drug delivery 2020, 17, 898-910.
  16. Muheem, A.; Shakeel, F.; Warsi, M.H.; Jain, G.K.; Ahmad, F.J. A combinatorial statistical design approach to optimize the nanostructured cubosomal carrier system for oral delivery of ubidecarenone for management of doxorubicin-induced cardiotoxicity: In vitro-in vivo investigations. Journal of pharmaceutical sciences 2017, 106, 3050-3065.
  17. Al-Joufi, F.A.; Salem-Bekhit, M.M.; Taha, E.I.; Ibrahim, M.A.; Muharram, M.M.; Alshehri, S.; Ghoneim, M.M.; Shakeel, F. Enhancing ocular bioavailability of ciprofloxacin using colloidal lipid-based carrier for the management of post-surgical infection. Molecules (Basel, Switzerland) 2022, 27.
  18. Alanazi, F.K.; Lu, D.R.; Shakeel, F.; Haq, N. Density gradient separation of carborane-containing liposome from low density lipoprotein and detection by inductively coupled plasma spectrometry. Journal of liposome research 2014, 24, 53-58.
  19. Liu, G.; Zhao, X.; Zhang, Y.; Xu, J.; Xu, J.; Li, Y.; Min, H.; Shi, J.; Zhao, Y.; Wei, J., et al. Engineering biomimetic platesomes for ph-responsive drug delivery and enhanced antitumor activity. Adv Mater 2019, 31, e1900795.
  20. Javed, S.; Alshehri, S.; Shoaib, A.; Ahsan, W.; Sultan, M.H.; Alqahtani, S.S.; Kazi, M.; Shakeel, F. Chronicles of nanoerythrosomes: An erythrocyte-based biomimetic smart drug delivery system as a therapeutic and diagnostic tool in cancer therapy. Pharmaceutics 2021, 13.
  21. Pei, W.; Huang, B.; Chen, S.; Wang, L.; Xu, Y.; Niu, C. Platelet-mimicking drug delivery nanoparticles for enhanced chemo-photothermal therapy of breast cancer. Int J Nanomedicine 2020, 15, 10151-10167.
  22. Hu, Q.; Sun, W.; Qian, C.; Wang, C.; Bomba, H.N.; Gu, Z. Anticancer platelet-mimicking nanovehicles. Adv Mater 2015, 27, 7043-7050.
  23. Wang, H.; Wu, J.; Williams, G.R.; Fan, Q.; Niu, S.; Wu, J.; Xie, X.; Zhu, L.M. Platelet-membrane-biomimetic nanoparticles for targeted antitumor drug delivery. J Nanobiotechnology 2019, 17, 60.
  24. Liao, Y.; Zhang, Y.; Blum, N.T.; Lin, J.; Huang, P. Biomimetic hybrid membrane-based nanoplatforms: Synthesis, properties and biomedical applications. Nanoscale horizons 2020, 5, 1293-1302.
  25. Bang, K.H.; Na, Y.G.; Huh, H.W.; Hwang, S.J.; Kim, M.S.; Kim, M.; Lee, H.K.; Cho, C.W. The delivery strategy of paclitaxel nanostructured lipid carrier coated with platelet membrane. Cancers (Basel) 2019, 11, 807.
  26. Chen, Y.; Zhao, G.; Wang, S.; He, Y.; Han, S.; Du, C.; Li, S.; Fan, Z.; Wang, C.; Wang, J. Platelet-membrane-camouflaged bismuth sulfide nanorods for synergistic radio-photothermal therapy against cancer. Biomater Sci 2019, 7, 3450-3459.
  27. Luo, J.; Schmaus, J.; Cui, M.; Hörterer, E.; Wilk, U.; Höhn, M.; Däther, M.; Berger, S.; Benli-Hoppe, T.; Peng, L., et al. Hyaluronate sirna nanoparticles with positive charge display rapid attachment to tumor endothelium and penetration into tumors. J Control Release 2021, 329, 919-933.
  28. Kim, J.E.; Park, Y.J. High paclitaxel-loaded and tumor cell-targeting hyaluronan-coated nanoemulsions. Colloids Surf B Biointerfaces 2017, 150, 362-372.
  29. Ye, H.; Wang, K.; Wang, M.; Liu, R.; Song, H.; Li, N.; Lu, Q.; Zhang, W.; Du, Y.; Yang, W., et al. Bioinspired nanoplatelets for chemo-photothermal therapy of breast cancer metastasis inhibition. Biomaterials 2019, 206, 1-12.

Comment 9. Authors should briefly comment on the disadvantages of the studied systems. One whole paragraph should be added to let readers get better insight into the subject.

Response 9: Thank you very much for your precious advice. According to your suggestion, we have briefly commented on the disadvantages of the studied system in the last paragraph of conclusion section (page 10, lines 307-311) as follows:

Thus far, the anticancer effect of the emerging biomimetic nanomedicine has not been fully demonstrated in humans. In our work, CCM@PL was prepared using platelet membrane of rat for preliminary lab research, which is not suitable for human use. In the future, we will use human platelet membrane to improve this bioinspired formulation for further research.

Comment 10. Conclusion: The conclusion should be concise and to the point indicating the application of the work.

Response 10: Thank you for your valuable advice. Based on this work, CCM@PL has been speculated to be an encouraging application for tumor targeted therapy. According to your suggestion, we have revised the conclusion section and indicated the application of the work (page 10, lines 300-311) as follows:

The developed formulation CCM@PL showed prolonged circulation time and improved bioavailability. Moreover, PL exhibited good targeting capacity toward Huh7, SK-OV-3, and MDA-MB-231 cells. This might be attributed to the biomolecular binding between platelet P-selectin and cancerous CD44. In addition, CCM@PL exhibited remarkable cytotoxic effects on cancer cells. Therefore, CCM@PL has been speculated to be an encouraging application for tumor targeted therapy. Future studies are planned to confirm the in vivo targeting capacity of CCM@PL as well as in vivo anticancer effect.

Thus far, the anticancer effect of the emerging biomimetic nanomedicine has not been fully demonstrated in humans. In our work, CCM@PL was prepared using platelet membrane of rat for preliminary lab research, which is not suitable for human use. In the future, we will use human platelet membrane to improve this bioinspired formulation for further research.

Thank you for all the valuable and helpful comments and suggestions. We hope that our revised manuscript is now suitable for publication in Pharmaceutics.

Best regards,

Jianming Wu

Reviewer 2 Report

The authors addressed cancer therapy via P-selectin targeting using bioinspired platelet-like nano vector. Before acceptance of the paper, the author must address following concerns.

1) In Figure 1B, characterization of the CCM@PL were carried out using confocal laser scanning microscopy. The spatial resolution of CLSM is at most 200 nm, even if STED is used, so characterization using CLSM is not appropriate for CCM@PL with a diameter of 160 nm. Please explain a valid reason why this method of characterization is correct or withdraw the data from the manuscript.

2) In Figure 3a, why fluorescence signals from DiR in blood is not uniform? In solution or dispersion, the fluorescence should be uniform. In addition, the fluorescence from blood should be measured by fluorometer directly.

3) The author had better address whether current system work in vivo as targeting and therapeutics.

Author Response

Nov. 20, 2022

Dear Expert Reviewer,

Thank you very much for the prompt review process and excellent comments. We greatly appreciate the time and efforts which you have spent on it. We are submitting the revised manuscript entitled “Bioinspired Platelet-like Nanovector for Enhancing Cancer Therapy via P-selectin Targeting” (ID: pharmaceutics-1962608) to Pharmaceutics.

We have carefully considered your comments and suggestions, and addressed each of the concerns in response to the comments (see point by point response). We have revised the manuscripts based on your comments and carefully checked throughout the manuscript and corrected the language errors. Our point-by-point responses to the comments (in blue) are shown below (in red).

Comment 1. In Figure 1B, characterization of the CCM@PL were carried out using confocal laser scanning microscopy. The spatial resolution of CLSM is at most 200 nm, even if STED is used, so characterization using CLSM is not appropriate for CCM@PL with a diameter of 160 nm. Please explain a valid reason why this method of characterization is correct or withdraw the data from the manuscript.

Response 1: Thank you very much for your precious advice.

The characterization of CCM@PL was carried out using previously reported confocal laser scanning microscopy (CLSM) analysis (Liu. et al. Adv Mater 2019, Zhang. et al. J Nanobiotechnology 2021, He. et al. ACS Nano 2019). The cell membrane biomimetic lipid vesicles with size of less than 200 nm could be detected by CLSM in these previous reports. And our result was similar with these studies (Figure R1-R3). Probably because membrane biomimetic lipid vesicles are easy to aggregation (Corbo et al. Int J Nanomedicine 2016), the biomimetic lipid vesicle clusters may be observed by CLSM. We had learned the relevant principles of CLSM, nanoparticle with size of less than 200 nm cannot be detected by CLSM. Thus, we just saw the vesicle clusters in this work. Thanks a lot for you friendly suggestion, we had withdrawn the data from the revised manuscript to avoid ambiguity.

Figure R1 please find it in the  attachment word file.

Figure R1. Platelet membrane biomimetic liposome with a size of 122.6 nm was observed by confocal microscopy. The platelet membrane nanovesicles (PNV) and liposomes were, respectively, labeled with PKH26 (red) and PKH67 (green). Scale bar = 10 μm. (Liu. et al. Adv Mater 2019)

Figure R2 please find it in the  attachment word file.

Figure R2. Confocal fluorescent microscopy image of cancer cell membrane biomimetic liposome with a size of 195.2 nm (red = cancer cell membrane, green = liposomes; scale bar = 1 μm). (Zhang. et al. J Nanobiotechnology 2021)

Figure R3 please find it in the  attachment word file.

Figure R3. Confocal fluorescence images of a red blood cell membrane biomimetic liposome with a size of 110 nm. Red, lipid membrane; green, red blood cell membrane; scale bar = 1 μm. (He. et al. ACS Nano 2019)

References

Liu, G.; Zhao, X.; Zhang, Y.; Xu, J.; Xu, J.; Li, Y.; Min, H.; Shi, J.; Zhao, Y.; Wei, J., et al. Engineering biomimetic platesomes for pH-responsive drug delivery and enhanced antitumor activity. Adv Mater 2019, 31, e1900795.  (Liu. et al. Adv Mater 2019)

Zhang, W.; Gong, C.; Chen, Z.; Li, M.; Li, Y.; Gao, J. Tumor microenvironment-activated cancer cell membrane-liposome hybrid nanoparticle-mediated synergistic metabolic therapy and chemotherapy for non-small cell lung cancer. J Nanobiotechnology 2021, 19, 339.  (Zhang. et al. J Nanobiotechnology 2021)

He, Y.; Li, R.; Li, H.; Zhang, S.; Dai, W.; Wu, Q.; Jiang, L.; Zheng, Z.; Shen, S.; Chen, X., et al. Erythroliposomes: Integrated hybrid nanovesicles composed of erythrocyte membranes and artificial lipid membranes for pore-forming toxin clearance. ACS Nano 2019, 13, 4148-4159.  (He. et al. ACS Nano 2019)

Corbo, C.; Molinaro, R.; Taraballi, F.; Toledano Furman, N.E.; Sherman, M.B.; Parodi, A.; Salvatore, F.; Tasciotti, E. Effects of the protein corona on liposome-liposome and liposome-cell interactions. Int J Nanomedicine 2016, 11, 3049-3063.  (Corbo et al. Int J Nanomedicine 2016)

Comment 2. In Figure 3a, why fluorescence signals from DiR in blood is not uniform? In solution or dispersion, the fluorescence should be uniform. In addition, the fluorescence from blood should be measured by fluorometer directly.

Response 2: Thank you very much for your precious advice and rigorous thinking. In our work, the plasma samples were taken into the centrifugal tube for fluorescence imaging (Figure 3C). It is possible that the centrifuge tube shielded part of the fluorescence signal and some liquid stuck to the wall of the tube, thus the fluorescence signals were not uniform (Yan. et al. J Nanobiotechnology 2020, Li et al. J Control Release 2022). The fluorescence images obtained by image system in our work (Figure 3C) were similar with the previous reports (Liu. et al. Adv Mater 2019, Yan. et al. J Nanobiotechnology 2020, Li. et al. J Control Release 2022), as shown in Figures R4-R6. Meanwhile, the fluorescence intensities of plasma samples were measured by a microplate reader (VLBL00D0, Thermo Fisher, Waltham, USA) directly. The result showed that the fluorescence intensity from DiR@PL group was higher than that from the DiR group at 2, 3, 4, 6 and 8 h after administration, indicating the prolonged circulation of DiR@PL (Figure 3A). Despite utilizing different technologies, the result obtained by microplate reader was in line with the result obtained by imaging system. And this result obtained by microplate reader has been added in the revised manuscript (page 7, lines 251-254, page 8, lines 259-265).

Figure 3C please find it in the  attachment word file.

Figure 3C. Representative fluorescence images of plasma samples derived from rats treated with DiR or DiR@PL, repeatedly collected at the indicated time points post administration.

Figure R4 please find it in the  attachment word file.

Figure R4. Fluorescence images of blood samples derived from mice treated with various Cy5.5 labeled vesicles. (Liu. et al. Adv Mater 2019)

Figure R5 please find it in the  attachment word file.

Figure R5. Ex vivo imaging of the plasma after intravenous injection of DiD formulations. (Yan. et al. J Nanobiotechnology 2020)

Figure R6 please find it in the  attachment word file.

Figure R6. DiD-labled formulations in blood at 72 h post-injection. (Li. et al. J Control Release 2022)

Figure 3A please find it in the  attachment word file.

Figure 3A. Relative fluorescence intensities of plasma samples derived from rats treated with DiR or DiR@PL, detected by a microplate reader.

References

Yan, F.; Zhong, Z.; Wang, Y.; Feng, Y.; Mei, Z.; Li, H.; Chen, X.; Cai, L.; Li, C. Exosome-based biomimetic nanoparticles targeted to inflamed joints for enhanced treatment of rheumatoid arthritis. J Nanobiotechnology 2020, 18, 115.  (Yan. et al. J Nanobiotechnology 2020)

Li, H.; Feng, Y.; Zheng, X.; Jia, M.; Mei, Z.; Wang, Y.; Zhang, Z.; Zhou, M.; Li, C. M2-type exosomes nanoparticles for rheumatoid arthritis therapy via macrophage re-polarization. J Control Release 2022, 341, 16-30.  (Li. et al. J Control Release 2022)

Liu, G.; Zhao, X.; Zhang, Y.; Xu, J.; Xu, J.; Li, Y.; Min, H.; Shi, J.; Zhao, Y.; Wei, J., et al. Engineering biomimetic platesomes for pH-responsive drug delivery and enhanced antitumor activity. Adv Mater 2019, 31, e1900795.  (Liu. et al. Adv Mater 2019)

Comment 3. The author had better address whether current system work in vivo as targeting and therapeutics.

Response 3: Thank you very much for your valuable advice. In our work, the research of CCM@PL was in the preliminary experimental stage. And we are preparing to conduct relevant in vivo experiments such as in vivo targeting and therapeutics. Based on results in this work, CCM@PL has been speculated to be an encouraging application for tumor targeted therapy in vivo. Future studies are planned to confirm the in vivo targeting capacity of CCM@PL as well as in vivo anticancer effect. According to your suggestions, we have added this speculation regarding the application of CCM@PL for in vivo targeting and therapy in the conclusion section of revised manuscript (page 10, line 304-305).

Thank you for all the valuable and helpful comments and suggestions. We hope that our revised manuscript is now suitable for publication in Pharmaceutics.

Best regards,

Jianming Wu

Round 2

Reviewer 1 Report

The authors have revised the manuscript according to my concerns. The revised manuscript is suitable for publication in its present form.

Reviewer 2 Report

The author addressed my concern.